# Long-term Morphodynamics of a Schematic River Analysed with a Zero-dimensional, Two-reach, Two-grainsize Model

Mariateresa Franzoia<sup>1</sup>, Michael Nones<sup>2</sup>, Giampaolo Di Silvio<sup>1</sup>

- <sup>5</sup> <sup>1</sup>Department of Civil, Architectural and Environmental Engineering, University of Padua, via Loredan 20, 35100, Italy <sup>2</sup> Interdepartmental Centre for Industrial Research in Building and Construction - Fluid Dynamics Unit, University of Bologna, Italy, via del Lazzaretto 15/5, 40121 Bologna, Italy *Correspondence to*: Michael Nones (michael.nones@unibo.it)
- Abstract. At the basin scale, neglecting localized deviations caused by geological constraints like knick-points, alluvial rivers commonly exhibit a concave profile and a progressive sediment fining in the downstream direction. Although this configuration is, perforce, not in equilibrium, yet it usually shows a quasi-stationary behaviour at the very long (historical and geological) time-scale.

A zero-dimensional, two-reach, two-grainsize hydro-morphological model is presented and applied to a schematic river. The

15 description of the processes involved is done assuming several reasonable and verified simplifications, giving reason of the extremely slow evolution of many alluvial rivers and providing a quantitative approach to evaluate their 'response time'. Different from previous analytical formulations, the response time appears here to be affected, among others, by the granulometry of the sediment input.

Applications of the model to different geometries demonstrate that the present riverine morphology at the basin scale will persist practically stationary for extremely long periods of time (centuries and even millennia), if the climatic forcing remain

unchanged and no anthropogenic perturbations are introduced in the system.

# **1** Introduction

Since a long time, researchers argue about the overall configuration of alluvial rivers that, regardless of their size, present a tendentially concave profile accompanied by the fining of the bottom material in the downstream direction (see, among

- others, Richter, 1939; Shulits, 1941; Leopold and Maddock, 1953; Snow and Slingerland, 1987; Sinha and Parker, 1996; Gomez et al., 2001; Rice and Church, 2001; Ferrer-Boix et al., 2016). The presence of hard-rock formations may create discontinuities (i.e., knick-points, waterfalls and lakes) in this configuration (Church, 2002). However, neglecting the presence of such discontinuities, the bottom profile appears to be concave and the grainsize progressively become finer from upstream to downstream. Although the trend in fining is preserved in many river systems, the overall pattern of textural
- change is very complex (Troutman, 1980; Pizzuto, 1995; Gasparini et al., 1999; Surian, 2002). To explain this behaviour,

deviations from the usual concave profile have been attributed to geologic controls (Ouchi, 1985; Schumm et al., 2000), climatic history or tectonics of the river basin (Ouchi, 1985; Rice and Church 2001; Bishop et al., 2005), base-level lowering (Begin et al., 1981), or tributary inputs of coarse material (Miller, 1958; Seidl and Kirchner, 1994; Rice and Church, 1998). The commonly observed grainsize fining has been initially justified by the abrasion of particles transported by watercourses

- (Sternberg, 1875). This process, however, has been soon recognized as seldom relevant for most lithological materials, and the decrease of particle diameter has been rather attributed to a progressive granulometric selection, somehow associated to a flattering of the bottom slope. The river profile concavity and composition have been explained through the conceptual notion of a graded landform system, in which the slope declines downstream as an interplay between flow discharge, sediment load, bed grainsize and channel morphological characteristics (Kesseli, 1941; Mackin, 1948; Hack, 1957; Harmar
- and Clifford, 2007; Ferrer-Boix et al., 2016). In its basic form, such a behaviour is suggested by any sediment transport formula, as well as the 'equivalent' expression of the so-called 'Lane's balance' (Lane, 1955; Dust and Wohl, 2012), indicating that, in equilibrium conditions, coarser sediments should conform to a steeper slope. Nevertheless, a concave profile and a progressive fining as found in nature cannot represent real equilibrium conditions (i.e. a stationary configuration). Despite operative issues (Métivier et al., 2016), laboratory experiments show that, assuming stationary liquid
- and solid inputs, the profile will get flatter and flatter and the grainsize increasingly uniform, aiming to attain the only physically possible steady configuration: a uniform slope. Following these considerations, one should expect that the concavity and the fining displayed by alluvial watercourses represent a sort of quasi-equilibrium (i.e. quasi-stationary) conditions, at least at the historical scale (decades and centuries). Namely, a morphological configuration evolving at an extremely slow rate towards a theoretical asymptotic flat and uniform configuration, eventually reached if the boundary 20 conditions remain unchanged (Davis, 1899).

In the present paper, a two-reach, two-grainsize, zero-dimensional schematic model is applied (Franzoia, 2014), deduced by integrating over the two reaches a LUF (Local Uniform Flow) one-dimensional, hydro-morphodynamic model (Fasolato et al., 2009-2011; Nones, 2012; Nones et al., 2014). This model is an expansion of an existing one-reach, zero-dimensional schematic model, previously applied to define a 'response time' of fluvial systems (Di Silvio and Nones, 2014). That model,

- which assumed a uniform grainsize material and a single straight profile, indicated that the response time of the river is proportional to a sole morphometric parameter called 'filling time'. This represents the time required by the sediment discharge to fill the so-called 'filling volume', namely the volume defined by the total river surface multiplied by its average elevation with respect to the base level.
- On the one hand, the adoption of a two-grainsize and two-reach approach permits a more realistic representation of alluvial rivers. On the other hand, it has required the introduction of other morphometric parameters to completely describe the river configuration and its evolution, especially in terms of bottom fining and profile concavity. As will be seen, however, the filling time of the river remains the basic morphometric quantity affecting the response time. In the next section, the constitutive equations are presented and described both in dimensional and dimensionless terms. The model is then applied to six schematic case studies, with the aim to assess the relative contribution of the different parameters adopted. The results

highlight the reliability of the model in describing the long-term evolution of alluvial rivers, and provide opportune criteria to define the 'equilibrium' and the 'quasi-equilibrium' conditions of fluvial systems. Above all, the model gives reason of the present configuration of alluvial rivers of the world.

# 2. Mathematical modelling

5 In this section, the constitutive equations of the model are introduced and discussed. The 0-D model uses a formulation in dimensionless terms, aiming to pinpoint the relative importance of the few non-dimensional morphometric parameters controlling the evolution of any type of alluvial river.

# 2.1 Constitutive equations

To compare the morphodynamic behaviour of different alluvial rivers in a relatively simple way, and still retaining the essential features of each fluvial system, several simplifications have been introduced. First, the hydrographic basin surface is supposed to be integrated in its barycentre at the upstream end of the river channel, where water and sediment inputs from the basin slopes are concentrated (Franzoia, 2014). The absence of water and sediment feeding along the river, either by tributaries or by lateral inputs, implies a constant river width corresponding to a long-term constant discharge along the watercourse. Although this assumption may appear very tough, it is instrumental for an acceptable simplification of the

- 15 problem and represents the ordinary conditions of many laboratory experiments regarding sediment transport. The evolution of the bottom profile and grainsize composition of the river channel is described by the 1-D equations of water flow and sediment transport, written under the following hypotheses: instantaneous water flow propagation and Local Uniform Flow (LUF) conditions (Fasolato et al., 2009-2011; Nones, 2012). This latter hypothesis allows to simplify the description of the water flow, considering that the averaged energy line and the averaged water and bed profiles have the same slope.
- 20 Regarding the granulometric compositions of the river bottom, for the sake of simplicity only two representative grainsizes classes are here considered, characterising fine and coarse fractions of sediments, indicated with the subscripts f and c, respectively (Franzoia, 2014).

Considering well-developed subcritical and supercritical conditions, hydro-morphodynamics can be uncoupled (De Vriend et al., 1993; Juez et al., 2013). Therefore, the equations used for the simplified 1-D model are reduced to the sediment
continuity equation along the stream (Eq. 1), proposed by Exner (1920), and the mass balance for each grainsize fraction in the active layer (Eq. 2), developed by Hirano (1971).

$$B\frac{\partial H}{\partial t} = -\sum_{k=1}^{2} \frac{\partial P_{k}}{\partial x}$$

$$\partial \left(\beta_{k}\right) = \partial P_{k} \qquad \partial h$$
(1)

$$\delta B \frac{\langle k \rangle}{\partial t} = -\frac{\partial k}{\partial x} - \beta_k B \frac{\partial h}{\partial t}$$
(2)

where *B* indicates the active (transport) river width,  $P_k$  is the transport of the *k*-th fraction of sediments, *h* represents the bed elevation,  $\delta$  is the thickness of the bottom mixing layer,  $\beta_k$  indicates the percentage of the *k*-th fraction of the grainsize composition that varies during the erosion or deposition phases, having a single active layer.

A relationship of the Engelund-Hansen type (Armanini and Di Silvio, 1988; Nones, 2012) has been assumed to express the sediment transport  $P_k$ :

$$P_{k}(x,t) = \alpha_{EH} \frac{Q(x,t)^{m}}{B(x,t)^{p}} \frac{I(x,t)^{n}}{d_{k}^{q}} \beta_{k}(x,t) \varsigma_{k}(x,t)$$
(3)

where  $\alpha_{EH}$  is a proportionality coefficient, Q indicates the water flow discharge, I represents the bottom slope (coinciding with averaged energy line and water profile slopes),  $d_k$  is the diameter of the *k*-th fraction of sediments and  $\zeta_k$  indicates the hiding-exposure coefficient, which considers the interactions of different grainsize classes in the mixture (Egiazaroff, 1965;

Ribberink, 1987).

The exponent *m* generally ranges between 1.5 and 2.5 and is site-specific. This parameter should be calibrated for each river depending on the watercourse characteristics (Nones, 2012), while the other exponents *n*, *p* and *q* are proportional to it. As an example, adopting the Chézy formula to describe the hydraulics, one obtains n=m, p=(m-1) and q=3/2(m-1).

Two representative grainsizes classes (coarse and fine fractions) are assumed to describe the bottom composition, which is  $\beta$ 15 for fine sediments characterized by a diameter  $d_{f}$ , and  $(1-\beta)$  for the coarse one, which has a characteristic diameter  $d_c$ .

The total sediment transport is given by the sum  $P(t) = \sum P_k(t)$ . Namely:

$$P(x,t) = \alpha_{EH} \frac{Q(x,t)^m}{B(x,t)^p} \frac{I(x,t)^n}{d_{eq}(x,t)^q}$$
(4)

where  $d_{eq}$  is the equivalent diameter, expressed where  $d=d_f/d_c$  and s accounts for the hiding-exposure coefficient  $\zeta_k$ .

$$d_{eq}(x,t)^{q} = \frac{\beta(x,t)[(1/d)^{q-s} - 1] + 1}{d_{c}^{q}[\beta(x,t)(d-1) + 1]^{s}}$$
(5)

The composition of the total sediment transport related to the fine fraction  $\alpha = P_f(t)/P(t)$  (or  $1 - \alpha = P_c(t)/P(t)$  for the coarse grains) depends on the bottom composition  $\beta$ .

$$\alpha(x,t) = \frac{\beta(x,t)(1/d)^{q-s}}{\beta(x,t)[(1/d)^{q-s} - 1] + 1}$$
(6)

Collecting the coefficients and the morphological quantities in an overall parameter *M* called morphodynamic conveyance, a sediment rating curve can be obtained (e.g., Asselman, 2000; Horowitz, 2003; Franzoia, 2014):

$$P(\tau) = \alpha_{EH} \frac{1}{B^p} \frac{I^n}{d_{eq}} Q(\tau)^m = M Q(\tau)^m$$
 (7)

where all the quantities, except  $Q(\tau)^m$ , are supposed to be virtually constant at the (seasonal or shorter) time scale  $\tau$  (Franzoia, 2014).

Averaging the Eq. (7) over the short-term time  $\tau$ , it is possible to assume that the morphodynamic conveyance M(t) is a function of the time-averaged parameters  $\overline{I}(t)$  and  $\beta(t)$ , variable at the long-time scale  $t \gg \tau$  (Franzoia, 2014).

$$M(t) = \alpha_{EH} \frac{\overline{I}(t)^{n}}{\overline{B}^{p}} \frac{\overline{\beta}(t) [(1/d)^{q-s} - 1] + 1}{d_{c}^{q} [\overline{\beta}(t)(d-1) + 1]^{s}} =$$

$$\alpha_{EH} \frac{\overline{I}(t)^{n}}{\overline{B}^{p}} \frac{c_{1}[\overline{\beta}(t)]}{d_{c}^{q}} = C \cdot \overline{I}(t)^{n} c_{1}[\overline{\beta}(t)]$$
(8)

where C incorporates all the invariable parameters and  $c_1(t)$  is an implicit function of  $\overline{\beta}(t)$  and d.

$$c_{1}\left[\overline{\beta}(t)\right] = \left[\frac{d_{c}}{d_{eq}\left[\overline{\beta}(t)\right]}\right]^{q} = \frac{\overline{\beta}(t)\left[\left(1/d\right)^{q-s} - 1\right] + 1}{\left[\overline{\beta}(t)(d-1) + 1\right]^{s}}$$
(9)

Note that, at the long-time scale t, the constant quantity B in Eq. (1) becomes the valley width  $B_V$  occupied by the river during its wandering, while the averaged quantity  $\overline{B}$  in Eq. (8) is still the active river width, conveying water and sediments.

#### 10 2.2 Zero-dimensional, two-reach model

The stream is schematized in two connected uniform-slope channels, with no lateral confluences (Figure 1), aiming to represent alluvial rivers using a 0-D model. From a physical point of view, regardless possible local discontinuities, these two parts represent the highland reach with steeper slopes and coarser grainsize, and the lowland reach, characterized by milder slopes and finer sediments. The upstream reach (indicated with the subscript U) starts in the barycentre of the

15 watershed feeding solid and liquid supply to the system, while the downstream reach (subscript D) ends at the river outlet and conventionally starts at a distance from the outlet corresponding to the largest tributary's confluence (conjunction point). The higher reach has a constant valley width  $B_U$  generally narrower than the constant valley width  $B_D$  of the lower part, because of geology and natural constrictions such as rock banks.

Figure 1: Scheme of the 0-D model: a) longitudinal and b) planimetric view.

In Figure 1, L indicates the total river length, which remains constant during the simulation and is given by the sum of  $L_U$ 5 (constant length of the upstream reach) and  $L_D$  (constant length of the downstream reach). H(t) is the river relief, changing in time t towards  $H_{\infty}$  that is reached at the equilibrium time  $t_{\infty}$ . G(t) is the sediment input from the entire watershed, concentrated at the upstream end (watershed barycenter), while  $P_U(t)$  and  $P_D(t)$  indicate the sediment transport flowing out the upstream and downstream reach, respectively.

Ordinary differential equations describing the behaviour of the two-grainsize, two-reach system can be obtained integrating 10 Eqs. (1) and (2) over the upstream  $L_U$  and over downstream  $L_D$  lengths, accounting for the two grainsize classes:

$$\begin{cases} \frac{d\overline{I_{U}}}{dt} = \frac{2}{\overline{B_{U}}L_{U}^{2}} \left[ G(t) - P_{U}(t) - \frac{2L_{U}}{L_{D}} \frac{B_{U}}{B_{D}} \left[ P_{U}(t) - P_{D}(t) \right] \right] \\ \frac{d\overline{\beta_{U}}}{dt} = \frac{1}{\overline{B_{U}}L_{U}} \overline{\delta} \left[ \left( \alpha_{G}(t) - \overline{\beta_{U}}(t) \right) G(t) - \left( \overline{\alpha_{U}}(t) - \overline{\beta_{U}}(t) \right) P_{U}(t) \right] \\ P_{U}(t) = M_{U}(t)Q(t)^{m} \\ M_{U}(t) = Cost\overline{I_{U}}(t)^{n}c_{1} \left[ \overline{\beta_{U}}(t) \right] \\ \frac{d\overline{I_{D}}}{dt} = \frac{2}{\overline{B_{D}}L_{D}^{2}} \left[ P_{U}(t) - P_{D}(t) \right] \\ \frac{d\overline{\beta_{D}}}{dt} = \frac{1}{\overline{B_{D}}L_{D}} \overline{\delta} \left[ \left( \overline{\alpha_{U}}(t) - \overline{\beta_{D}}(t) \right) P_{U}(t) - \left( \overline{\alpha_{D}}(t) - \beta_{D}(t) \right) P_{D}(t) \right] \\ P_{D}(t) = M_{D}(t)Q(t)^{m} \\ M_{D}(t) = Cost\overline{I_{D}}(t)^{m}c_{1} \left[ \overline{\beta_{D}}(t) \right] \end{cases}$$
(10)

If the initial conditions of the system at the time t=0 are prescribed, the integration of (10) provides the temporal evolution of the averaged quantities  $\overline{I}(t)$ ,  $\overline{P}(t)$ ,  $\overline{\rho}(t)$  and M(t).

#### 2.3 Non-dimensional formulation

5 Geometrical and hydraulic quantities in Eq. (10) describe the evolution of alluvial watercourses, but, to reduce the complexity of the description, unknown variables and given parameters into a non-dimensional formulation will be used thereafter. The dimensionless variables reported in Table 1 are expressed as the relative deviations from their respective equilibrium values, reached at the equilibrium time  $t_{\infty}$ .

| dimensional parameters              | dimensionless parameters |  |  |
|-------------------------------------|--------------------------|--|--|
| $\beta_{U}, \beta_{D}$              | bu, bd                   |  |  |
| $\overline{I_U}$ , $\overline{I_D}$ | iu, id                   |  |  |
| $P_U$ , $P_D$                       | <i>pU</i> , <i>pD</i>    |  |  |
| G                                   | g                        |  |  |
| $Q^m$                               | $q_w^m$                  |  |  |
| $M_U$ , $M_D$                       | Morph,U, Morph,D         |  |  |

10 Table 1. Dimensional and dimensionless morphometric characteristics of the river.

Note that the equilibrium values of sediment discharges  $G(t \rightarrow \infty)$ ,  $P_U(t \rightarrow \infty)$ ,  $P_D(t \rightarrow \infty)$ , grainsize compositions  $\alpha_G(t \rightarrow \infty) = \alpha_U(t \rightarrow \infty) = \alpha_D(t \rightarrow \infty)$  and  $\overline{\beta_U}(t \rightarrow \infty) = \overline{\beta_D}(t \rightarrow \infty)$ , and bottom slopes  $I_U(t \rightarrow \infty) = I_D(t \rightarrow \infty)$  coincide. Namely, bed slope, sediment discharge and grainsize composition become increasingly uniform during time.

$$5 \quad \begin{cases} l_r = \frac{L_U}{L} \\ 1 - l_r = \frac{L_D}{L} \\ b_r = \frac{\overline{B_U}}{\overline{B_D}} \end{cases}$$

$$(11)$$

Defining the respective equilibrium values as  $G_{\infty}$ ,  $\alpha_{\infty}$ ,  $\beta_{\infty}$ ,  $I_{\infty}$  and introducing the geometrical dimensionless ratios (11), Eq. (10) can be rewritten in a dimensionless form:

$$\begin{cases} \frac{di_{U}}{dt^{*}} = \left[1 + \frac{1}{b_{r}} \left(\frac{1 - l_{r}}{l_{r}}\right)^{2} + 2\frac{1 - l_{r}}{l_{r}}\right] \\ \left[g(t^{*}) - p_{U}(t^{*}) - 2\frac{l_{r}}{1 - l_{r}} b_{r}(p_{U}(t^{*}) - p_{D}(t^{*}))\right] \\ \frac{d\beta_{U}}{dt^{*}} = \frac{I_{x}L}{\overline{\delta}} \left[\frac{l_{r}}{2} + \frac{1}{2b_{r}} \frac{(1 - l_{r})^{2}}{l_{r}} + (1 - l_{r})\right] \\ \left[\left(\alpha_{G}(t^{*}) - \beta_{L}(t^{*})\right)(g(t^{*}) + 1\right) - \left(\alpha_{U}(t^{*}) - \beta_{U}(t^{*})\right)(p_{U}(t^{*}) + 1)\right] \\ p_{U}(t^{*}) + 1 = \left[m_{orph,U}(t^{*}) + 1\right]\left[q_{w}(t^{*})^{m} + 1\right) \\ m_{orph,U}(t^{*}) + 1 = \left[i_{U}(t^{*}) + 1\right]^{n} \frac{c_{1}(\beta_{U}(t^{*}))}{c_{2}(\beta_{\infty})} \\ \frac{di_{D}}{dt^{*}} = \left[1 + b_{r}\left(\frac{l_{r}}{1 - l_{r}}\right)^{2} + 2\frac{l_{r}}{1 - l_{r}}\right] \left[p_{U}(t^{*}) - p_{D}(t^{*})\right] \\ \frac{d\beta_{D}}{dt^{*}} = \frac{I_{x}L}{\overline{\delta}} \left[\frac{b_{r}}{2} \frac{l_{r}^{2}}{(1 - l_{r})} + \frac{(1 - l_{r})}{2} + b_{r}l_{r}\right] \\ \left[\left(\alpha_{U}(t^{*}) - \beta_{D}(t^{*})\right)(p_{U}(t^{*}) + 1) - \left(\alpha_{D}(t^{*}) - \beta_{D}(t^{*})\right)(p_{D}(t^{*}) + 1)\right] \\ m_{orph,D}(t^{*}) + 1 = \left[l_{D}(t^{*}) + 1\right]^{n} \frac{c_{1}(\beta_{D}(t^{*})}{c_{2}(\beta_{\infty})} \end{cases}$$

$$(12)$$

where the non-dimensional time  $t^*=t/T_{fill}$  is defined through the filling time  $T_{fill}=V_{\infty}/G_{\infty}$ , already introduced by Di Silvio and 10 Nones (2014) in their one-reach analytical model. Here, the filling volume  $V_{\infty}$  is expressed as:

$$V_{\infty} = \frac{I_{\infty}\overline{B_U}L_U^2}{2} + \frac{I_{\infty}\overline{B_D}L_D^2}{2} + \overline{B_U}L_UI_{\infty}L_D$$
(13)

Given initial and boundary conditions, Eq. (12) provide the evolution of the non-dimensional variables  $i_U(t^*)$ ,  $i_D(t^*)$ ,  $\beta_U(t^*)$ ,  $\beta_U(t^*)$ ,  $p_U(t^*)$ ,  $p_D(t^*)$ ,  $m_{orph,U}(t^*)$  and  $m_{orph,D}(t^*)$ ; namely slope, bottom composition, sediment transport and morphodynamic conveyance of, respectively, upstream and downstream reach.

- An inspection of (12) indicates that a relatively small number of independent non-dimensional quantities are necessary to depict the behaviour of alluvial rivers of any size and configuration. Besides the filling volume  $V_{\infty}$ , two other geometrical parameters contribute in describing the river valley: the length ratio  $l_r = L_U/L$  between upstream and total reach, and the width ratio  $b_r = B_U/B_D$  between upstream and downstream valley widths (11). The composition of the material fed into the river, represented by the composition  $\alpha_G$ , together with its amount  $G_{\infty}$  incorporated in the filling time, determine the elevation of
- the equilibrium relief  $H_{\infty}$  of the river.

#### 3. Results and discussion

To calibrate the model showing the effects of the dimensionless parameters on the long-term river evolution, six numerical tests have been carried out, as indicated in Table 2. This analysis was performed modifying the river-basin length ratio  $l_r$ , the river-valley width ratio  $b_r$  and the sediment production composition  $\alpha_G$  (via the modification of the corresponding

- equilibrium slope  $I_{\infty}$ ). Tests 2 and 4 have an upstream part  $L_U$  longer that the downstream one  $L_D$ , similarly to alluvial rivers that flow from mountains to the sea, while the other tests assume that the lowland part is longer that the upstream one. In Tests 3 and 4 the river is free to wandering, especially in its lowland part, as the valley widths  $B_U$  and  $B_D$  are larger than the active river width *B*, different from other tests that assumed a confined river bed (width ratios  $B_U/B=B_D/B=1$ ). Regarding the sediment input, three configurations were analysed, spanning from finer ( $\alpha_G=0.995$ ) to coarser ( $\alpha_G=0.900$ ) sediments. The
- starting sediment compositions drive to differences in the equilibrium slope  $I_{\infty}$ .

| Test | $lpha_G$ | $I_{\infty}$ | Lu/L  | $L_D/L$   | BU/B  | B <sub>D</sub> /B |  |
|------|----------|--------------|-------|-----------|-------|-------------------|--|
|      |          |              | $l_r$ | $(1-l_r)$ | $b_r$ |                   |  |
| 1    | 0.950    | 1.55 10-3    | 0.17  | 0.83      | 1     | 1                 |  |
| 2    | 0.950    | 1.55 10-3    | 0.83  | 0.17      | 1     | 1                 |  |
| 3    | 0.950    | 1.55 10-3    | 0.17  | 0.83      | 20    | 200               |  |
| 4    | 0.950    | 1.55 10-3    | 0.83  | 0.17      | 20    | 200               |  |
| 5    | 0.900    | 2.05 10-3    | 0.17  | 0.83      | 1     | 1                 |  |

6

0.83

1

1

0.17

Earth Surface

Dynamics

Discussions

To evaluate the validity of the model in describing different behaviours, all the tests have been implemented integrating the Eq. (11) with the same initial conditions, which simulate a schematic, instantaneous orogeny. A flat horizontal bottom characterized by a uniformly distributed coarse bed composition and a constant input of material were imposed (namely, at

5 the time  $t^*=0$ ,  $i_U=i_D=0$  and  $\beta_U=\beta_D=0$ ; while for  $t^*>0$ , *G* and  $\alpha_G$  were kept constant and equal to their respective equilibrium values). The evolution of the river profile proceeded from the initial flat profile at elevation zero to the asymptotic uniform-slope configuration, as indicated in Figure 1.

Two morphometric time-depending quantities are significant in describing the evolution of the river:

0.78 10-3

- profile concavity (relative difference of reach slopes)

0.995

10 
$$X(t^*) = \frac{I_U(t^*) - I_D(t^*)}{I_x} = i_U - i_D$$
 (14)

- river fining (relative difference of bottom composition)

$$\Phi(t^*) = \frac{\beta_D(t^*) - \beta_U(t^*)}{\beta_{\infty}} = b_D - b_U$$
(15)

The temporal changes of concavity  $X(t^*)$  and fining  $\Phi(t^*)$  for the six tests are reported in Figure 2.