# Peer review of "Long-term Morphodynamics of a Schematic River Analysed with a Zero-dimensional, Two-reach, Two-grainsize Model"

_Earth Surface Dynamics, 2017_

## Referee Comment (RC1) · Anonymous Referee #1 · 28 Mar 2017

Summary: This manuscript develops a simplified physically based model of the evolution of a river profile with two distinct reaches and grain size fractions. A subset of the potential parameter space for the model is explored through the presentation of six morphologic parameter combinations.

General comments: The manuscript would benefit greatly by connecting with the rich literature on landscape and river evolution models and showing why the current approach adds something new to this discussion. As it currently stands it is difficult to discern what the novelty of this contribution to the literature is.

Currently, the model demonstrates that the long-term equilibrium state is one of no concavity (one slope) and thus suggests that because rivers are commonly concave

they must not be at an equilibrium state. However, I remained unconvinced that the model actually represents the mechanics or even physical representation of an actual river and rather seems to approximate the profile and set up of a sediment feed flume, which does indeed capture aspects of a river profile but leaves out important details. In that the final state of the model captures the expected behavior of a two size fraction feed flume (see discussion in Parker and Wilcock, 1995). A key missing ingredient in connecting with natural upland or mountain catchments in the current set up is the lack of a connection between the profile and the supply of sediment. In the current set up (with constant sediment supply) the river profile must always evolve to a state where it can transport the input sediment and thus needs to evolve to a state where it can always transport the coarse sediment which indicates that the final equilibrium slope always has to equal the upstream segments final slope. Such a constraint limits the types of landscapes that this model could be applied to (i.e. landscapes that approximate flumes). I strongly encourage the authors to consider how their model assumptions have dictated their final equilibrium states and that these states may not be applicable to many natural landscapes. Or I encourage the authors to consider narrowing the scope of the conclusions and discussion to reflect the conditions and limitations within their model framework.

This manuscript would benefit from the use of an english language editing service.

Specific comments:

P. 2 Ln. 7 - Flattening rather than flattering, unless the river has indeed paid you a complement.

P. 2 Ln. 5-7 - The authors would benefit from reading all and working into the introduction some of the following papers on the sorting of particles and how it links to the slope and long profile of a river (Seal et al. 1997; Paola and Seal 1995; Cui et al., 1996; Fedele and Paola, 2007; and especially the following two Paola et al., 1992a; and Paola et al. 1992b). Many of the questions posed in this introduction are addressed in these

works.

P. 2 Ln. 14 - Be sure to update the citation to the Metivier et al. 2016 work (the work is now published).

P. 2 Ln. 14-20 - These lines seem a little out of place with the narrative, especially the ideas of Davis at the end of the paragraph. The end morphologic state of an alluvial river under constant flow would be one where sediment transport has ceased and the channel is everywhere at threshold (Lacy's Law, from Metivier et al., 2016, cited earlier in the text), not something with no slope (flat).

P. 5 Ln. 17 - Why are rock banks and geology used as constraints for a narrower upper segment when the model apriori assumes no such natural complexities. It would seem fine given the assumptions made earlier to just state the model set up (in figure 1) as the initial conditions rather than justifying the conditions via externalities the authors have sought to avoid.

P. 8 Ln. 2 - It is not clear why at equilibrium the bed grain sizes and slopes should be equal for both reaches. This would seem to preclude the system to be able to sort the coarse and fine sediments, which seemed to be the reason behind having a two reach two grain size model.

P. 10 Ln. 8 - It would make more sense to write '... in describing the evolution of the model' rather then the 'river'.

Figure 2. This figure takes awhile to fully comprehend and would benefit from a new figure showing the evolution of the river profile at different time points. Additionally it would help to change the x-axis label from t* to log(t*).

P. 11 Ln. 27-29 - Don't the longterm implications of the model show that there is no concavity ($i\_U = i\_D$)? How does this model relate to natural rivers that are concave? It is stated later in the conclusions that an implication would be that because rivers are concave they must not be at their equilibrium state, but this seems to be a bold

conclusion based on the many simplifications present. Please consider an alternative possibility that the longterm implications of the equations are not indicative of an actual river, but may represent a sediment feed flume where under constant discharge capable of transporting all size classes (as in the model here) the resulting end state will have a single slope (to a first order) and one bed surface composition (see discussion in Parker and Wilcock, 1993). Consider also the work of Guerit et al., (2014), particularly figure 9 where their 1D river (alluvial fan) which would very closely approximate a 1 reach 1 size fraction version of this model still displays concavity at its final state and the physics of the argument suggest that the concavity is related to sediment transport.

P. 11 Ln. 31 - Figure 2 does not seem to support the assertion that Tfill is the most important morphometric parameter determining the longterm river evolution. In Figure 2 the various runs do not converge at Tfill=1 (log(Tfill) =0) or display any change in behavior (no inflection points or maximal/minimal values). From Figure 2 it seems that the various runs converge on their final equilibrium behavior near Tfill =10. Some discussion should be added here as to why it takes 10 times the time needed to fill the space to achieve the equilibrium values. If Tfill is a better predictor of other important parameters, those parameters should probably be added to an additional figure that supports the assertion of its importance.

References cited Cui, Y., Paola, C. and Parker, G.: Numerical simulation of aggradation and downstream fining, Journal of Hydraulic Research, 34(2), 185–204, doi:10.1080/00221689609498496, 1996.

Fedele, J. J. and Paola, C.: Similarity solutions for fluvial sediment fining by selective deposition, J. Geophys. Res.-Earth Surf., 112(F2), F02038, doi:10.1029/2005JF000409, 2007.

Guerit, L., Métivier, F., Devauchelle, O., Lajeunesse, E. and Barrier, L.: Laboratory alluvial fans in one dimension, Phys. Rev. E, 90(2), 22203, doi:10.1103/PhysRevE.90.022203, 2014.

Paola, C. and Seal, R.: Grain-Size Patchiness as a Cause of Selective Deposition and Downstream Fining, Water Resour. Res., 31(5), 1395–1407, doi:10.1029/94WR02975, 1995.

Paola, C., Parker, G., Seal, R., Sinha, S., Southard, J. and Wilcock, P.: Downstream Fining by Selective Deposition in a Laboratory Flume, Science, 258(5089), 1757–1760, doi:10.1126/science.258.5089.1757, 1992a.

Paola, C., Heller, P. L. and Angevine, C. L.: The large-scale dynamics of grain-size variation in alluvial basins, 1: Theory, Basin Research, 4, 73–90, 1992b.

Parker, G. and Wilcock, P. R.: Sediment Feed and Recirculating Flumes: Fundamental Difference, Journal of Hydraulic Engineering, 119(11), 1192–1204, doi:10.1061/(ASCE)0733-9429(1993)119:11(1192), 1993.

Parker, G. and Wilcock, P. R.: Closure to "Sediment Feed and Recirculating Flumes: Fundamental Difference" by Gary Parker and Peter R. Wilcock, Journal of Hydraulic Engineering, 121(3), 293–294, doi:10.1061/(ASCE)0733-9429(1995)121:3(293), 1995.

Seal, R., Paola, C., Parker, G., Southard, J. and Wilcock, P.: Experiments on Downstream Fining of Gravel: I. Narrow-Channel Runs, Journal of Hydraulic Engineering, 123(10), 874–884, doi:10.1061/(ASCE)0733-9429(1997)123:10(874), 1997.
* * *
**ESurfD**

---

## Author Comment (AC1) · 11 Apr 2017

We are sincerely grateful to the Reviewer # 1 for the valuable comments on our work and the rich literature suggested on this topic. Waiting for the comments of other reviewers, we would like to respond meanwhile to the Comments of Reviewer # 1, which require more details about the novelties of our paper and the validity of the river representation. Additional replies will be given, following the observation of the other reviewers, with the final revision of the text.

GENERAL COMMENTS

Scope of our paper is integrating the ample literature covering different aspects of

downstream fining along rivers with a simple compact model interpreting the evolution of both river profile and bottom grainsize composition at the very large space- and time-scale. Indeed, the variety of scales in river systems plays an important role in our model and is probably the most interesting novelty of the paper.

Morphological changes in rivers take place at very short (flood event), short (seasonal), intermediate (historical), long and very long (geological) scale. However, only at the shortest scale laboratory and field experiments on morphodynamics are feasible, while at intermediate scale numerical investigations can be somehow carried out by means of conventional hydro-morphologic models. At the very large scale, by contrast, morphological simulations require substantial simplifications of the conventional models, mostly based on various types of averaging and aggregating operations. We will see in the following the simplifications adopted in our model.

Another aspect discussed in our paper, clearly connected with the concept of scale, is the "equilibrium configuration" of a river system or subsystem: namely the steady state of all the relevant hydraulic and morphological quantities, eventually attained in response to sufficiently persistent steady boundary conditions acting on the river (sub) system.

In our model the considered space-scale is that of the entire river system (watershed) and the (consequent) time-scale is the geological scale. This does not mean that the river sub-systems at a smaller scale (e.g. tributary, reach, river width, river depth, bed forms, mixing layer etc.) are totally irrelevant for the evolution of the river system, but that their effects are implicitly accounted for by algebraic equations, under the hypothesis of equilibrium conditions with their respective higher subsystem.

At the basin and geological scale, the boundary conditions of the river system are represented by the long-term water and sediment input from the basin slopes, and by the sea level at the downstream end. These quantities are supposed to be steady, even if the controlling climate may experience some oscillations at the geological scale.

In any case, even for middle size hydrographic basins, the response time of the river system proves to be too long for attaining the equilibrium conditions corresponding to these steady boundary conditions. In fact, the model confirms that, for modern rivers, the existence of a "quasi-equilibrium configuration", which evolves at an extremely slow rate but is quite different and very far from the expected equilibrium conditions.

It is interesting to note that the quasi-equilibrium configuration invariably presents a very plausible profile concavity and downstream grainsize fining, more or less accentuated depending on the basin characteristics. Moreover, owing to its long persistence, the quasi-equilibrium configuration constitutes a stable reference for assessing the effects on the river system of anthropogenic action or climatic change at a shorter (historical) time-scale.

The validity if the system representation adopted in our model is discussed in the attached file.

Please also note the supplement to this comment:
http://www.earth-surf-dynam-discuss.net/esurf-2017-7/esurf-2017-7-AC1-supplement.pdf

**Supplement:**

**Validity of the river system representation**

The conventional morphodynamic model of a river reach at short and intermediate scale, usually consists of the standard one-dimensional partial differential equations (de St. Venant, Exner and Hirano with one or more active layers) and the annexed semi-empirical algebraic equations that describe the water flow and the sediment motion in the stream and in the bottom, complemented by the necessary boundary conditions. To simulate the evolution of a river system, this set of equations should in principle be applied to the hydrographic network of the river, together with a time-dependent and space-dependent predictor of water and sediment production (by surface- and mass erosion) entering the streams from the basin slopes.

To reduce the computational effort, the partial differential equations are often simplified (Fasolato et al., 2011) and the model schematization limited to the largest branches of the hydrographic network. Moreover, if the analysis is devoted to long-term simulations, the partial differential equations are properly averaged (to include algebraically the effects of the shorter time-scales) over pluriannual time-steps, as mentioned in the previous section. Despite all these expedients for reducing the numerical complexity, however, the computational effort remains largely high. Even more important, similar models are still too much detailed to highlight the salient aspects of the long-term morphodynamics for broad categories of river basins. For this reason, we decided to go further with the process of aggregating and averaging the basin model and yet preserving the essential peculiarities of each river system.

First, following a procedure widely shared by both hydraulic engineers and geomorphologists, we aggregated the tree-shaped hydrographic network into a number of reaches connected in series and representing the different parts of the main watercourse. A second important simplification consists in integrating the 1D partial differential equations over the length of each reach. In this way, these partial differential equations are transformed into ordinary differential equations, while the physical description of the basin is expressed in terms of "concentrated" (0D) parameters, much more concise and manageable than the "distributed" (1D) parameters. The limit-case of aggregating the entire hydrographic network into one single reach of uniform width fed from upstream corresponds, in fact, to the case of the "sediment fed flume" mentioned by the Reviewer. Although very synthetic, the sediment fed flume scheme has been utilized by several researchers (often assuming uniform grain size material) for analysing the reaction of rivers to different types of perturbation. These applications (both with 1D and 0D approach) show the relevance of the overall parameter "morphological dispersion" closely related to the "response time" of the river (Paola et al, 1992 b: Castelltort and van Driessche, 2003; Gupta, 2007; Di Silvio and Nones, 2014 etc.).

The 0D single-reach single-grainsize scheme, however, is, by definition, unable to investigate the specific aspects we are interested in: the general tendency of rivers to display a concave profile and a grainsize fining in the downstream direction. To analyze this behaviour, adding the minimum complication to the single reach scheme, we applied a 0D two-reach two-grainsize scheme. This approach, distinguishing an upland and a lowland segment of the basin, permits in fact to portray a vast variety of river systems. As indicated in Section 2.2 of the paper, the partition between the two segments is made in such a way as to mimic the basic structure of the hydrographic network and to put into account the planimetric characteristics of each reach (length and width) of the valley, inferred from any geographical data base.

The long-term evolution of the system is provided by eq. 10 of the paper, when initial and boundary conditions are prescribed. If the boundary conditions (equivalent waterflow $Q(t)$, sediment input $G(t)$ and input grainsize composition $\alpha_G(t)$ remain constant, the system will evolve extremely slowly from a flat hypothetical initial condition (orogeny) towards the already mentioned

"equilibrium conditions", as shown in the Figure reported here.

[Figure]

The evolving profiles shown in the Figure are expressed in non-dimensional terms as they represent the solution of eq.12, namely the non-dimensional formulation of eq.10. Eq.12 indicates that the long-term evolution of the system towards a uniform slope $I_\infty$ and a uniform bed composition $\beta_\infty$ can be described by only *five independent non-dimensional parameters* ($br$, $lr$, $\alpha_G$, $I_\infty$ and $\beta_\infty$), which incorporate all the relevant morphometric quantities describing the river system and its boundary conditions.
Namely:
- $B$, constant river width (due to the assumption of constant $Q$) measured near the month;
- $B_U$ and $B_D$, averaged valley widths (allowing the river wandering), respectively for the upstream and downstream reach;
- $L_U$ and $L_D$, length of the main watercourse (properly defined) for the upstream and downstream reach;
- $Q$, constant (equivalent) waterflow, accounting for the hydrological variations at flood and seasonal scale;
- $G$, constant long-term sediment input (surface and mass erosion) from the basin slopes;
- $\alpha_G$, grainsize sediment composition of the sediment input.

Note that, by assuming any reasonable sediment transport formula for a sediment mixture (e.g. eq. 3 of the paper), the prescribed boundary conditions $Q$, $G$ and $\alpha_G$ univocally define the equilibrium slope $I_\infty$ and the equilibrium bed composition $\beta_\infty$ of the river. In this way, the non-dimensional quantities $I_\infty$ and $\beta_\infty$ appearing in eq. 12 constitute a proxy of the prescribed equivalent waterflow $Q$, not explicitly included in the same equations.
The five independent non-dimensional parameters, moreover, identify together another important quantity; namely, the "filling time" $T_{fill}= V_\infty/G$, which is the only fundamental parameter in the one-reach one-grainsize scheme.
In the present model, the filling time (although defined in a somewhat more complex way, see eq.13 of the paper) is still the scaling quantity of the long-term time $t$, but is not anymore *proportional to the response time* of the river system as in the case of the one-reach one-grainsize scheme. In the two-reach, two-grainsize model, the five parameters $br$, $lr$, $\alpha_G$, $I_\infty$ and $\beta_\infty$ define indirectly the rapidity of reaction and, more in general, the long-term behaviour of the system.

The evolution of a certain river system, characterized by the five non-dimensional parameters, is represented in the Figure by a series of profiles at different values of the non-dimensional time $t^*$. A similar representation may also be obtained for the bottom composition of the two reaches. From these representations, however, one cannot immediately perceive how evolve the quantities we are more interested in, namely the overall *concavity* and *fining* of the system.

The model tests reported in the paper show that both the non-dimensional concavity $X(t^*)$ and fining $\Phi(t^*)$ (by definition equal to zero for $t^*=0$ and $t^*=\infty$) tend to present a persistent maximum value within an intermediate range of $t^*$ ($0.1<t^*<10$), depending upon the five parameters characterizing the river basin. In any case, it seems reasonable to presume that maximum values for $X(t^*)$ and $\Phi(t^*)$ correspond more or less to the "quasi-equilibrium configuration" of the present rivers. We plan therefore to verify the validity of this hypothesis by simulating the long-time evolution at geological scale of several real rivers, in a wide range of morphoclimatic conditions, i.e. size, hydrology and lithology of the basin.

**References**

Castelltort, S., Van Den Driessche, J.: How plausible are high-frequency sediment supply-driven cycles in the stratigraphic record? Sedimentary Geology 157, 3-13, 2003.

Di Silvio, G., and Nones, M.: Morphodynamic reaction of a schematic river to sediment input changes: Analytical approaches. Geomorphology, 215, 74-82, doi: 10.1016/j.geomorph.2013.05.021, 2014.

Fasolato, G., Ronco, P., Langendoen, E. J., and Di Silvio, G.: Validity of Uniform Flow Hypothesis in One-Dimensional Morphodynamics Models. Journal of Hydraulic Engineering, 137(2), 183–195, doi: 10.1061/(ASCE)HY.1943-7900.0000291, 2011.

Gupta, A.: Large Rivers: Geomorphology and Management. eds. A. Gupta, Wiley, 2007

Paola, C., Heller, P. L. and Angevine, C. L.: The large-scale dynamics of grain-size variation in alluvial basins, 1: Theory, Basin Research, 4, 73–90, 1992b.

---

## Referee Comment (RC2) · Anonymous Referee #2 · 3 May 2017

In this submission, Franzoia et al. present a mathematical framework within which it may be possible to analyse the evolution of experimental and natural alluvial rivers. They invoke a suite a simple, analytically tractable hard assumptions about a river system to perform the analysis, of which the principle are: 1. The system is chute-like, 2. The system is driven by constant, time-invariant inputs of water & sediment; 3. Mechanical abrasion of the transported sediment is negligible; 4. A local uniform flow assumption, i.e., that the bed slope is the energy slope of the flow, and 5. That the system is fully alluvial, i.e., the Exner equation always applies. The derived set of analytical expressions is then used to assert that the system described tends to a planar profile, and that this response occurs over a well-defined timescale related to

the sediment volume flux and the initial (and boundary) conditions. The manuscript concludes with a sensitivity analysis performed as a set of six numerical case studies examining the effects of modifying various of the driving parameters and boundary conditions. They suggest that all six share a common response pattern of a rapid initial transient response, a drawn out, quasi-equilibrium period during which the profile is concave and fines downstream, then a slow approach to the linear condition and zero fining.

I struggled with the assumptions underlying this manuscript, and so cannot recommend it for publication in its current form. Because the approach is purely analytical, the authors are forced to make strong assumptions about process and form in order to render the equation system analytically tractable. Each of these in isolation could be regarded as questionable when applied to a real river system; together, they feel like an unacceptable stretch if describing a real river (see below). These concerns would be lessened if the framework were explicitly applied to an experimental setup (only), where at least the validity of the various assumptions in the experiment could be more easily tested or ensured by the method. However, this leads to the main, overarching weakness of the text as presented: it remains totally ungrounded in physical reality, be that the field or experiment. Without this element in the work presented, a reviewer can critique only the hard assumptions in an abstract sense, and on this basis it has proven challenging for the authors to defend this framework as it stands.

With that in mind, I would specifically question the following:

1. Early in the introduction, the authors dismiss out of hand the role of abrasive fining in alluvial systems (p2 lns4-7). This statement is – bizarrely – purely asserted, and lacks any citation to support what is definitely a controversial interpretation of the literature on this. My reading is that the relative contributions of abrasion and size-selective sorting remains very much an open question – see e.g. various work by Attal, Lavé and coauthors, including a piece in last week's (27/4) Nature by Dingle and colleagues. This assumption may be justifiable for an experiment, but the paper seeks explicitly to

link the framework presented to field examples of rivers.

2. P2, lns 12-16: The comments about necessary disequilibrium in concave natural rivers. This is a very hard assumption, and is model driven (i.e., it is a hypothesis emerging from assumptions made in deriving this and similar frameworks; it does not follow immediately from physical reality). It, too, is uncited. I appreciate that this is the conceptual framework that these authors wish to push as important both in this work and the sequence which precedes it, but here it is being presented as an obvious and unarguable observation, rather than the somewhat circular assertion it appears to be. This impression is aggravated by the absolutist language around the claim (e.g., "cannot" at ln 13, "perforce" at ln 12 in the abstract). This language needs at the least to be shifted out of the abstract and intro and into the Discussion. A wider discussion of whether this is a plausible hypothesis for real rivers is probably appropriate, if the manuscript retains its focus on real rivers.

As a related issue, I note that today (3/5) a manuscript appeared in the Int. J Sed Res by two of the same authors (10.1016/j.ijsrc.2017.04.002) which seems extremely similar in its approach to this manuscript (sharing the bulk of its equations), but the conclusions seem quite different. In particular, in that work, the authors indicate that the similar 0-D, two reach, two grainsize model tends towards a stable, downstream fining, concave profile. Subtle differences are present (e.g. a quick scan reveals differences in handling of channel width, grain size distributions, and hiding functions), but given the shared authorship, difference in resulting conclusions, and very similar approach, I think a discussion of where and why differences appear between the manuscripts would be appropriate.

3. The influence of stationary water and sediment inputs. This I think gets at the heart of the field vs. experimental issue underlying the paper. Many experiments are run under these conditions, and so arguably the presented model would be justified narrowly for such experiments. However, this is manifestly not a valid assumption for real rivers, and indeed, has been invoked within a variety of papers (including a clutch cited in the

introduction) as key in stabilizing and driving field-measured concavities. The lack of tributary input is also troubling in this model for the same reason. This is very much a hard assumption; it's not possible to say a priori under the analysis presented here what effect relaxing it would have. However, we can hypothesise based on previous studies, and indeed first principles, that time variant discharges must have an effect. In particular, they must alter assumptions around time invariant hiding functions and other parameters (e.g., Equ 3), and surely affect the rationale presented at the top of p 5, where the authors explicitly argue for a fully equilibrium system at a "seasonal or shorter" timescale.

4. A related issue is the invoking of the LUF assumption. I agree that this is an elegant and useful minimal complexity approximation of the SWEs, but it remains just that – an approximation. Under the assumption of constant discharges it seems fully applicable (as the system would seek hydrologic steady state, as described), but if we were to relax that assumption, it's not clear to me that this would remain appropriate. This condition is known to suppress instabilities in fluvial systems (see e.g. work by Smith & Bretherton), and is perhaps the driving assumption behind the tendency of these models to seek a perfectly linear profile. But, as I say, who knows until the assumptions are relaxed and/or they are shown to be meaningful in reality.

A number of more minor issues came up during my read through. These essentially fall into three broad classes:

1. For a manuscript with so much detailed mathematics, the authors could be considerably more generous in talking the reader through the derivations and how various observations made arise from the math. For example, hiding-exposure coefficients could be a vital part of this story, but they are not explained in any detail, nor is their formulation as a constant term (? – if not, what is the justification of its disappearance into constants in later equations?) really justified. Another example might be the key result that the profile tends to a constant gradient (e.g. line 1 p 8), which appears to be made by observation from a suite of 8 differential equations. Another might be the

complex statement of how V_inf is defined at ln 10 p 8, which is extremely opaque – until one recalls the much clearer statement of what V_inf actually "is", drawn from a previous paper, presented pages back in the introduction. There are many examples. Please be less terse. 2. The English used can sometimes be convoluted to the point of inhibiting understanding. Most frustratingly, this seems to be worst in the abstract (e.g., "perforce", "giving reason of", but it is pervasive also. Much of the language also seems unnecessarily jargon-y (though technically accurate), especially sitting alongside the dense mathematics (e.g., "granulometry" where "grain size" would serve; "barycentre"; "quasi-equilibrium"). Plain English and well defined technical terms would help a lot here. 3. From my background, many of the terms deployed in the derivation are not using the Greek or Latin symbolology I would usually expect (e.g., slope is I not S; concavity is X not theta; sediment transport rate is P not Q_s, etc etc). If the authors have good reason for these formulations then fair enough, but if they are not following a standard scheme then they should at least consider moving to the more geomorphically typical symbology for clarity for the reader. Also, as a side note, I see that some of the symbol definitions are imprecise (e.g., P_bar is "averaged sediment transport", when this is more precisely a time- and reach-averaged sediment discharge). Please check them, as this can be very frustrating when trying to follow dense mathematics.

In summary, the key issue I have with this manuscript is that it contains a number of hard assumptions that cannot be uncontroversially justified when applied to either natural or experimental rivers. This could to a large degree be remedied by adding material to the manuscript explicitly making these comparisons, either experimental or natural, but I don't think the manuscript can stand without this. I imagine this would be readily possible in the case of a comparison to experiments, but not straightforward for real landscapes.

---

## Editor Comment (EC1) · JM Turowski (Editor) · 3 May 2017

Dear authors,

As you can see, we have now received two reviews for the manuscript. I think that both reviewers give many clear, relevant comments that need to be addressed in detail. In particular, the authors need to address the queries about the fundamental assumptions in the model set up (flume-like vs. field cases) and the range of applicability of the model.

Currently, the manuscript misses a discussion that puts the material into the context of existing literature. There are numerous models that predict concave-up river profiles –

the authors should look at downstream-hydraulic geometry and rational regime models; the seminal papers of Parker (e.g., J. Hydr. Eng. 1979) may provide a starting point for a literature research. I would like to see detailed explanations of where the model assumptions made by the authors differ from models in the literature, and a discussion of why they get different results from a process perspective. This could be followed by a short discussion of what kinds of landscape may feature the type of rivers described by the model, as suggested by the reviewers.

I also ask the authors to keep methods, results and interpretations separate. In particular, the results section should be separated from the discussion.

Best, Jens Turowski

---

## Author Comment (AC2) · 5 Jun 2017

We are sincerely grateful to Reviewers #1 for the valuable comments on our work and the rich literature suggested on this topic. Following such comments we revise the entire manuscript, and an additional discussion is reported here.

page 2, line 14-20: As specified in the revised Introduction of our manuscript, in our study we consider an alluvial river having a constant long-term flow and sediment input, and we search for its final morphological state. p. 5, l. 17: The model uses two diverse widths for the valley and the river. The different widths ($B_U$ and $B_D$) of the river valley (excluding the constant active river-width B) are not imposed constraints, but are

possible descriptors of the considered river system, basically controlled by external conditions (including geology), assumed here to remain constant during the profile and grainsize evolution. p. 8, l. 2: As visible in Eq. (10), setting equal to zero the time derivatives (equilibrium conditions), grainsize and slope, as well as sediment transport and transport composition, result equal for both reaches and compatible with the (constant) water and sediment input. p. 11, l. 27-29: One of the main hypothesis of the model is that a river network may be aggregated in a single watercourse, similar to a water- and sediment-fed flume. The water and sediment input is concentrated at the upstream end and the 'active' width of the river is constant. With the assumed initial conditions, the model develops an increasing concavity (and fining) that initially persists and later tends to disappear as the profile approaches the asymptotic equilibrium conditions (t→∞). This is what occurs also in the Guerit et al. (2014) experiments, carried out with a constant width of the flume. In the case of a 2-D alluvial cone, the geometrical widening of the cone circumference in the downstream direction would have very likely permitted a concave profile also at the asymptotic equilibrium conditions. It should be noted, however, that a progressively wider 'active' width along the cone is hardly realistic, as the stream will tend to wander all over the cone surface, keeping a relatively constant active width. p. 11, l. 31: The filling time Tfill is the time required by the sediment input G to fill the space comprised between the river bottom in equilibrium conditions and the base elevation. As there is always a considerable output from the river, the time to reach the equilibrium condition is necessarily very much longer than Tfill. In Fig. 3 of the present paper is shown that this occurs, in fact, for t* about 10, namely after a time near ten times Tfill, with some variations depending on the morphometric characteristics of the river basin. Comparable values of t* to reach the equilibrium conditions have been obtained with the 0-D one-reach, one-grainsize model (Di Silvio and Nones, 2014). However, what results most important is that – well before reaching the final equilibrium conditions – the model attains the maximum values for both the concavity and the fining. The high values of these parameters are quite persistent (quasi-stationary or quasi-equilibrium conditions), and may reasonably

**ESurfD**
represent the present configuration of real rivers. Note that, while Tfill is still a fundamental parameter inasmuch as it scales the morphodynamic time of the river evolution, other morphometric parameters (listed in Table 2) are also relevant, especially as far as the quasi-equilibrium configuration is concerned.

References: Di Silvio, G., and Nones, M.: Morphodynamic reaction of a schematic river to sediment input changes: Analytical approaches. Geomorphology, 215, 74-82, doi: 10.1016/j.geomorph.2013.05.021, 2014. Guerit, L., Métivier, F., Devauchelle, O., Lajeunesse, E. and Barrier, L.: Laboratory alluvial fans in one dimension, Phys. Rev. E, 90(2), 22203, doi:10.1103/PhysRevE.90.022203, 2014.

---

## Author Comment (AC3) · 5 Jun 2017

We are grateful to Reviewers #2 for the valuable comments, which contribute to improve the quality of our paper. Several doubts were tackled along the manuscript, and we report here an additional discussion of our model.

point 3: The cascade of time (and space) scales coexisting in river hydro-morphodynamics is indeed a relevant issue. Following the well tested procedure of subsequent simplifications for the Navier-Stokes equations, the problem is generally solved by opportune averaging of the Exner and Hirano equations, as it has been done for Eq. (7) and the following ones. Whenever the time-scales are sufficiently different

(as in the case of seasonal and geological scales), the averaged non-stationary terms at the shorter time scale ïĄť can be correctly incorporated in the stationary terms at the longer scale t using simple algebraic expressions. p. 4: Navier-Stokes equations are often simplified also by spatial averaging operations: for example, from 3-D to 2-D, 1-D and 0-D formulations. In particular, the 1-D equations (normally called de St. Venant eq.) may be further simplified into the LUF formulation if the length of the averaged river reach (morphological box) is sufficiently long, depending upon the river size and the Froude number of the current. This criterion has exactly been tested by relaxing the LUF assumptions (Fasolato et al., 2011).

References: Fasolato, G., Ronco, P., Langendoen, E. J., and Di Silvio, G.: Validity of Uniform Flow Hypothesis in One-Dimensional Morphodynamics Models. Journal of Hydraulic Engineering, 137(2), 183–195, doi: 10.1061/(ASCE)HY.1943-7900.0000291, 2011.